# Medication adherence and quality of life among geriatric patients: Insights from a hospital-based cross-sectional study in India

**Umaima Farheen Khaiser[1], Rokeya Sultana[1]\*, Ranajit Das[2], Saeed G. Alzahrani[3], Shahabe Saquib[4], Shaheen Shamsuddin[5], Mohammad Fareed[6]**

1 Department of Pharmacognosy, Yenepoya Pharmacy College and Research Centre, Yenepoya (Deemed to be University), Mangalore, Karnataka, India, 2 Division of Data Analytics Bioinformatics and Structural Biology, Yenepoya Research Centre, Yenepoya (Deemed to be University), Mangalore, Karnataka, India, 3 Department of Family and Community Medicine, College of Medicine, Al-Imam Mohammad Ibn Saud Islamic University (IMSIU), Riyadh, Kingdom of Saudi Arabia, 4 Department of Periodontics and Community Dental Sciences, College of Dentistry, King Khalid University, Abha, Kingdom of Saudi Arabia, 5 Department of Orthodontics, College of Dentistry, King Khalid University, Abha, Kingdom of Saudi Arabia, 6 Department of Environmental Health and Clinical Epidemiology, Saveetha Institute of Medical and Technical Sciences (SIMATS), Center for Global Health Research, Saveetha Medical College and Hospital, Chennai, India

\* rokeya009ster@gmail.com

**Data Availability Statement:** All relevant data are within the manuscript and its Supporting information files.

## Abstract

### Background

Understanding the factors that influence medication adherence and the multidimensional aspects of quality of life in the elderly is of paramount importance in enhancing their overall well-being. Since geriatric patients usually suffer from multiple morbidities due to their declining age, the adherence towards their medications plays a very crucial role in their quality of life.

### Methodology

This cross-sectional study explores the intricate relationship between medication adherence and quality of life among 310 elderly patients at a single medical college and hospital. Participants completed the Morisky Medication Adherence Scale (MMAS-8) to assess medication adherence and the World Health Organization Quality of Life-BREF (WHOQOL-BREF) questionnaire, which comprises four domains (physical health, psychological health, social relationships, and environment) to evaluate quality of life. Statistical analyses, including correlations, paired t-tests, ANOVA, and Backward Multiple Linear Regression, were employed to examine the relationships and differences among variables.

### Results

The findings indicate varying levels of medication adherence among participants, with a significant proportion exhibiting medium adherence (47.1%) and highlighting the need for interventions to address challenges in medication adherence among the elderly population. Notably, gender emerged as a significant factor influencing quality of life, with males

**Funding:** The author(s) received no specific funding for this work.

**Competing interests:** The authors have declared that no competing interests exist.

reporting higher satisfaction across all domains compared to females. Medication adherence exhibited a significant correlation with the social relationships domain (DOM3) of the WHOQOL-BREF, underlining the importance of adherence in fostering positive social interactions.

## Conclusion

Our study revealed a significant association between medication adherence (MMAS- 8) and the quality of life (WHOQOL-BREF) among elderly patients. We also observed noteworthy gender differences in quality-of-life perceptions. It emphasizes the need for tailored interventions that consider medication adherence issues to enhance the overall quality of life among this vulnerable population.

## Introduction

Population ageing is considered to be one of the most significant societal transformations [1]. The geriatric population has had a twofold increase since the year 1980 [2]. According to findings from the 2001 census, it was noted that there were around 77 million individuals categorized as elderly in the population. Subsequently, in 2011, this figure experienced an increase, reaching a total of 96 million. Projections indicate that by the year 2050, the elderly population is anticipated to escalate significantly, reaching an estimated 301 million individuals [2,3]. The primary health concerns faced by the geriatric population encompass non-communicable diseases (NCDs), like cardiovascular disease (CVD), diabetes mellitus (DM), chronic lung disease, hypertension, and cancer [3]. The continuing increase in the older demographic, along with a significant rise in the availability of prescription medications and adherence to treatment regimens, has presented a complex and demanding concern [2,4]. The elderly population has a heightened susceptibility to therapeutic non-adherence due to their increased morbidity and mortality rates, as well as the presence of cognitive and social impairments that impede proper medicine utilization [5].

The concept of quality of life is a complex and comprehensive construct that encompasses a range of dimensions related to individuals' well-being. These dimensions include physical health, mental health, functional abilities, social relationships, and general happiness with life [6,7]. The preservation of an ideal quality of life holds significant relevance for geriatric adults, as it directly influences their level of independence, capacity to engage in everyday activities, and emotional well-being [8]. The presence of chronic diseases and their corresponding symptoms can have a substantial negative impact on the overall quality of life among elderly individuals, necessitating the utmost importance of ensuring disease management using medication adherence [9].

Medication adherence pertains to the degree to which individuals adhere to the advice provided by their healthcare providers with regards to the time, dosage, and frequency of prescribed medication [10]. The elderly demographic faces unique difficulties in adhering to prescription regimens, leading to inconsistent medication adherence [11]. Difficulties in comprehending and sticking to intricate medication instructions can arise from various factors, including cognitive decline, memory problems, sensory limitations, and the coexistence of comorbidities [12]. Consequently, the rates of adherence among older persons exhibit significant variation, which may result in inadequate management of diseases and a decline in overall

quality of life [13]. Medication adherence and its impact on the quality of life among geriatric populations have garnered significant attention in recent years [14,15]. The ageing demographic is experiencing a notable increase globally, leading to a greater prevalence of chronic diseases and a subsequent rise in medication utilization among older adults [16]. The prevalence of medication adherence among the elderly varies between 40% and 75% across different settings [2]. Numerous reasons have been linked to non-adherence, including certain attributes of the medications prescribed, such as unpleasant reactions, pharmaceutical costs, inadequate instructions, and polypharmacy [17]. However, the effectiveness of these medications in enhancing the quality of life among geriatric individuals is closely tied to their adherence to prescribed regimens [18]. Medication non-adherence can have profound consequences, including increased healthcare costs, worsened disease outcomes, and diminished quality of life [19].

The relationship between medication adherence and quality of life in elderly adults is complex and mutually influential [20]. Adherence to recommended drug regimens has been shown to result in greater disease management, improved symptom control, and overall better health outcomes [21]. In the given context, a positive cycle is generated whereby enhanced medication adherence leads to enhanced health outcomes, subsequently contributing to an elevated standard of living [22]. In contrast, inadequate adherence to drug regimens can result in the advancement of diseases, exacerbation of symptoms, heightened rates of hospitalization, and a decline in overall quality of life [23]. There are various determinants that exert an influence on medication adherence and quality of life in the geriatric population. These determinants include socioeconomic status, healthcare accessibility, social support systems, cognitive functioning, and patient-provider communication [24,25]. Economic limitations can restrict the availability of medications, thereby impeding adherence [26]. Strong social support can encourage positive medication-taking behaviors, while cognitive decline can impede an individual's capacity to effectively manage medication schedules [27]. Therefore, establishing effective communication between healthcare providers and older patients is of paramount importance in addressing these determinants and facilitating both medication adherence and overall quality of life [26].

The ageing population is a global phenomenon and understanding how medication adherence affects the quality of life in this demographic is vital for healthcare systems worldwide [28,29]. Exploring the nexus between medication adherence and quality of life can offer insights into optimizing treatment strategies, improving health outcomes, and effectively utilizing healthcare resources. This study aims to evaluate the association between medication adherence and the quality of life among elderly patients with chronic diseases in Yenepoya Medical College and Hospital, Deralakatte, Mangalore, India.

## Methodology

### Setting and study design

A hospital based descriptive, cross-sectional study was conducted for elderly patients aged 60 years and above. The present study was carried out at Yenepoya Medical College and Hospital, located in Derelakatte, Mangalore, Karnataka, India. The hospital is a facility with a capacity of 1250 beds, serving an average of 650 out-patients and 300 in patients per day with a population of roughly 44,882 people.

### Sample size calculation

The sample size required for the present study was estimated via a sample-size calculator accessible on the website, www.raosoft.com The sample-size calculator facilitates the creation of a

representative sample by mitigating selection bias through a 5% margin of error, operating within a 95% confidence interval, and assuming a 50% response distribution. These parameters help minimize the skewness of the sample size and enable the determination of the maximum feasible sample size. A sample size of around 310 elderly patients, aged 60 years and above were selected.

## Screening of the patients (both in-patient & out-patient)

A simple random sampling technique was applied. Informed consent forms were given to the patients, and patients who were willing to participate had signed the consent form and were recruited depending upon the inclusion and exclusion criteria and then their demographic details. Medical history and medication history data were collected as per the data collection form. Informed consent was given to each patient and taken their signatures and documented.

**Inclusion criteria**: Patients aged 60 years and above who are receiving multi-drug therapy, defined as taking five or more medications for out-patients and nine or more medications for in-patients.

**Exclusion criteria**:

- Patients who are critically ill.

- Patients with cognitive impairment or mental retardation.

- Patients who are unwilling to participate in the study.

## Research ethics and permissions

The study received ethics approvals from the Yenepoya Medical College & Hospital Research Ethics Committee-1 (YEC-1), date of approval 01-06-2022.

## Data collection instrument

An epidemiological questionnaire was developed with the aim of investigating sociodemographic factors, including age and gender, as well as personal medical history. The WHO-QOL-BREF questionnaire is extensively employed as a generic instrument for assessing quality of life [30,31]. The questionnaire includes four domains, the components of each domain are mentioned in Table 1. WHOQOL-BREF scale contains a total of 26 items: items 3–26 represent four domains ("Physical Health" (DOM 1)—7 items (Pain and discomfort; Dependence on medicinal substances and medical aids; Energy and fatigue; Mobility; Sleep; Activities of daily living; Work capacity),"Psychological Health" (DOM 2)—6 items (Positive feelings; Spirituality/personal beliefs; Thinking, learning, memory and concentration; Bodily image and appearance; Self- esteem; Negative feelings), "Social Relationships" (DOM 3)—3 items (Personal relationships; Sexual activity; Social support), "Environment" (DOM 4)—8 items (Freedom, physical safety, and Security; Physical environment (pollution/noise/traffic/ climate); Financial resources; Opportunities for acquiring new information and skills; Participation in and opportunities for recreation/leisure activities; Home environment; Health and social care: accessibility and quality; Transport) [32]. The patients were questioned regarding how they had evaluated their health, quality of life, and other aspects of their lives over the previous two weeks. The responses to each questions are provided on a scale of 1 to 5, with 1 signifying the least agreement and 5 signifying the greatest agreement with the stated proposition. The domain score is calculated using the average score of the items in each domain. The total

**Table 1. Different domains for quality of life and their components according to WHO-QOL-BREF protocol.**

| Domain | Components within domains |
| --- | --- |
| 1. Physical health (DOM 1) | a. Activities of daily living<br>b. Dependence on medicinal substances and medical facilities<br>c. Energy and fatigue<br>d. Mobility<br>e. Pain and discomfort<br>f. Sleep and rest<br>g. Work Capacity |
| 2. Psychological (DOM 2) | a. Bodily image and appearance<br>b. Negative feelings<br>c. Positive feelings<br>d. Self-esteem<br>e. Spirituality / Religion / Personal beliefs<br>f. Thinking, learning, memory, and concentration |
| 3. Social relationships (DOM 3) | a. Personal relationships<br>b. Social support<br>c. Sexual activity |
| 4. Environment (DOM 4) | a. Financial resources<br>b. Freedom, physical safety, and security<br>c. Health and social care: accessibility and quality<br>d. Home environment<br>e. Opportunities for acquiring new information and skills.<br>f. Participation in and opportunities for recreation / leisure activities<br>g. Physical environment (pollution / noise / traffic / climate)<br>h. Transport |

of the results of items is represented by results on domains. A higher total of points indicates a higher standard of living in a certain domain.

The evaluation of adherence to the prescribed medication was conducted by administering an eight-item structured questionnaire to the patients, following the adaptation of the MMAS-8 (Morisky 8-item Medication Adherence Scale) for the specific context. The MMAS-8 is a widely used self-reported questionnaire designed to assess medication adherence. It consists of eight items, each with a binary response format (yes/no) or Likert scale. The scoring of MMAS-8 involves summing the responses to these items, with higher scores indicating greater adherence. Specifically, a score of 8 indicates high adherence, scores between 6 and 7 indicate medium adherence and scores below 6 indicate low adherence.

## Data collection process

The informed consent for participation in the study was obtained from the study participants. They have been informed about the process and importance of the study and have been assured that their details shall not be disclosed during and after the time of the study. The participants who had not provided consent were excluded from the study. The Ethics Committee-1 (YEC-1) had approved the consent procedure of the study. Following the acquisition of informed consent, the study participants were questioned at several locations including the out-patient department, and in-patient wards (general medicine ward, geriatric ward, and surgery wards) from 10th September 2022 to 5th August 2023. The data was gathered about the socio-demographic characteristics of the patients encompassing age, gender, IP/OP, medical and medication history by the utilization of a standardized questionnaire. The principal investigator along with the research assistant received training on the significance and utilization of the WHOQOL-BREF and the MMAS-8 instrument. They were also instructed for the method of data collection and were under the supervision of the public health professionals and

epidemiologists. Answering the questionnaire took approximately 20–25 minutes. Incomplete questionnaires were excluded from the analysis.

### Data analysis

The data was encoded and entered into an Excel spreadsheet and later transferred to SPSS. The data analysis was conducted using the SPSS version 26. The descriptive statistics for categorical variables were computed as frequency and percentages, for numerical variables mean and standard deviation. Correlation was done to determine the level of agreement between Medication adherence (MMAS-8) and four domains of WHOQOL-BREF. A paired t-test was done to compare the differences between the four domains of WHOQOL-BREF. To investigate the association between participants' sociodemographic characteristics and their WHO-QOL-BREF, an independent t-test (2 groups) and ANOVA test (>2 groups) were used. Finally, multiple linear regression (Backward method) was done to control the confounding effects. A probability value (p-value) of less than 0.05 was deemed to indicate statistical significance.

### Results

A total of 310 elderly patients were interviewed by the research team in the in-patient and out-patient department of Yenepoya Medical College and Hospital. Table 2 illustrates the socio-demographic characteristics of study participants. The study population had a mean age of 70.25±5.56 years. There were 122 female participants (39.4%), and 188 male participants (60.6%), who all completed the WHOQOL-BREF and MMAS-8 questionnaire. Most of the participants were 66–70 years (35.8%) of age group, followed by 71–75 years (24.2%), 60–65 years (22.3%), 76–80 years (13.9%), and 81 above years (3.9%) respectively. The majority of the study participants were from the In-patient ward (86.1%) and 13.9% were from the Out-patient department of the hospital. The majority of the subjects had medium adherence (47.1%) to the medication followed by low adherence (42.3%) and high adherence (10.6%) respectively.

**Table 2. Sociodemographic characteristics of study participants.**

| Variables | Frequency (n) | Percentage (%) |
|---|---|---|
| **Age group** | | |
| 60–65 years | 69 | 22.3 |
| 66–70 years | 111 | 35.8 |
| 71–75 years | 75 | 24.2 |
| 76–80 years | 43 | 13.9 |
| 81 and above | 12 | 3.9 |
| **Gender** | | |
| Male | 188 | 60.6 |
| Female | 122 | 39.4 |
| **In-Patient** | 267 | 86.1 |
| **Out-Patients** | 43 | 13.9 |
| **Adherence** | | |
| High | 33 | 10.6 |
| Medium | 146 | 47.1 |
| Low | 131 | 42.3 |

**Table 3. Test dataset descriptive statistics: WHOQOL-BREF.**

| Domains | N | Minimum | Maximum | QOL score Mean | Standard Deviation |
|---|---|---|---|---|---|
| Physical | 310 | 13 | 94 | 45.65 | 12.65 |
| Psychological | 310 | 13 | 94 | 46.64 | 10.47 |
| Social Relations | 310 | 19 | 100 | 63.67 | 11.65 |
| Environmental | 310 | 13 | 75 | 54.71 | 10.21 |

Table 3 shows descriptive statistics of QOL score concerning the four domains of WHO-QOL- BREF questionnaire response of the study participants. Table 4 illustrates the correlations between four domains of the WHOQOL-BREF score and MMAS-8. A statistically significant correlation was observed between medication adherence and DOM3 of the WHO-QOL-BREF score. Further, there is also a statistically significant correlation between all four domains of the WHOQOL-BREF score.

This study employed paired t-tests and ANOVA to assess the significant differences between the mean scores of various domain ratings. According to the data shown in Table 5, statistically significant differences were seen across all four distinct domains (pair 2, 3, 4, 5, and 6) of the WHOQOL-BREF score, except for the comparison between pair 1 (DOM1 and DOM2).

Among the several domains evaluated the domain with the highest satisfaction score was DOM3, with a mean value of 64.91. Conversely, the domain with the lowest satisfaction score was DOM1, with a mean value of 41.83. The mean score of four domains and total of WHO-QOL-BREF score according to age group, gender, IP/OP, and MMAS-8 are presented in Table 6. A statistically significant difference is observed about gender and Domain 3 of the WHOQOL-BREF score with p-value <0.05. The findings of Backward Multiple Linear Regression are presented in Table 7, indicating a significant association between the variable's "gender" and "MMAS-8" with the total WHOQOL score.

## Discussion

The topic of quality of life (QOL) holds significant importance when considering geriatric patients [33]. The state of an individual's health is widely recognized as a crucial determinant of their quality of life (QOL) since it significantly influences their degree of psychosocial

**Table 4. Correlation coefficients in medication adherence (MMAS-8) and four domains of WHOQOL-BREF.**

| | | Medication Adherence | DOM1 | DOM2 | DOM3 | DOM4 |
|---|---|---|---|---|---|---|
| **Medication Adherence** | Correlation Coefficient | 1 | 0.011 | -0.053 | 0.107 | -0.038 |
| | Sig. (2-tailed) | | 0.853 | 0.349 | **0.050** | 0.506 |
| **DOM1** | Correlation Coefficient | 0.011 | 1 | 0.551** | 0.237** | 0.289** |
| | Sig. (2-tailed) | 0.853 | | **<0.001** | **<0.001** | **<0.001** |
| **DOM2** | Correlation Coefficient | -0.053 | 0.551** | 1 | 0.210** | 0.306** |
| | Sig. (2-tailed) | 0.349 | **<0.001** | | **<0.001** | **<0.001** |
| **DOM3** | Correlation Coefficient | 0.107 | 0.237** | 0.210** | 1 | 0.076 |
| | Sig. (2-tailed) | **0.050** | **<0.001** | **<0.001** | | 0.181 |
| **DOM4** | Correlation Coefficient | -0.038 | 0.289** | 0.306** | 0.076 | 1 |
| | Sig. (2-tailed) | **0.506** | **<0.001** | **<0.001** | 0.181 | |

**. Correlation is significant at the 0.01 level (2-tailed).

**Table 5. Paired t-test for the four domains of WHOQOL-BREF.**

| | Paired Differences | | | | | | |
| | Mean | SD | 95% CI of the difference | | t test | df | Sig (2 tailed) |
|---|---|---|---|---|---|---|---|
| | | | Lower | Upper | | | |
| **PAIR 1** DOM1–DOM2 | -0.99 | 11.12 | -2.23 | 0.24 | -1.57 | 309 | 0.117 |
| **PAIR 2** DOM1–DOM3 | -18.02 | 15.04 | -19.70 | -16.34 | -21.09 | 309 | **<0.001** |
| **PAIR 3** DOM1–DOM4 | -9.06 | 13.77 | -10.60 | -7.52 | -11.59 | 309 | **<0.001** |
| **PAIR 4** DOM2–DOM3 | -17.02 | 13.93 | -18.58 | -15.47 | -21.51 | 309 | **<0.001** |
| **PAIR 5** DOM2–DOM4 | -8.07 | 12.18 | -9.43 | -6.71 | -11.66 | 309 | **<0.001** |
| **PAIR 6** DOM 3–DOM4 | 8.95 | 14.90 | 7.28 | 10.62 | 10.58 | 309 | **<0.001** |

Significance p-value<0.05; SD-Standard deviation; df–degree of freedom; CI- confidence interval.

functioning [34,35]. There has been a growing focus in the literature on the health-related quality of life of individuals with chronic diseases. This is due to the recognition that QOL can significantly impact an individual's perspective on their condition and their approach to treatment, including their level of medication adherence [36,37].

**Table 6. Comparison of WHOQOL-BREF transformed scores in four domains according to age-group, gender IP/OP, and medication adherence.**

| Parameters | N | Physical Health | | Psychological | | Social Relations | | Environ mental | |
|---|---|---|---|---|---|---|---|---|---|
| | | Mean | SD | Mean | SD | Mean | SD | Mean | SD |
| **Age-group** | | | | | | | | | |
| 60–65 | 69 | 45.69 | 13.17 | 46.59 | 11.77 | 63.31 | 10.88 | 54.95 | 10.72 |
| 66–70 | 111 | 45.99 | 10.66 | 47.56 | 9.57 | 64.34 | 11.76 | 56.00 | 9.49 |
| 71–75 | 75 | 46.49 | 13.90 | 45.73 | 10.37 | 63.66 | 10.84 | 53.29 | 10.64 |
| 76–80 | 43 | 44.30 | 14.15 | 45.09 | 10.99 | 62.27 | 14.41 | 52.58 | 10.63 |
| 81 and above | 12 | 41.83 | 13.88 | 49.66 | 9.28 | 64.58 | 10.17 | 58.00 | 7.78 |
| p-value | | 0.739 | | 0.499 | | 0.893 | | 0.171 | |
| **Gender** | | | | | | | | | |
| Male | 188 | 46.17 | 13.11 | 47.18 | 10.33 | 64.91 | 11.04 | 55.00 | 9.76 |
| Female | 122 | 44.85 | 11.90 | 45.18 | 10.66 | 61.76 | 12.34 | 54.28 | 10.89 |
| p-value | | 0.371 | | 0.259 | | **0.020*** | | 0.549 | |
| **OP/IP** | | | | | | | | | |
| Op | 43 | 45.38 | 12.52 | 46.33 | 10.12 | 63.86 | 11.60 | 54.36 | 10.07 |
| Ip | 267 | 47.32 | 13.08 | 48.55 | 12.35 | 62.48 | 12.07 | 56.90 | 10.86 |
| p-value | | 0.351 | | 0.197 | | 0.473 | | 0.130 | |
| **Medication Adherence** | | | | | | | | | |
| High | 33 | 42.36 | 11.03 | 45.84 | 13.62 | 60.06 | 14.57 | 54.87 | 9.09 |
| Medium | 146 | 46.95 | 12.48 | 47.78 | 10.13 | 63.56 | 11.90 | 55.18 | 10.66 |
| Low | 131 | 45.03 | 13.09 | 45.85 | 9.87 | 64.70 | 10.40 | 54.16 | 10.00 |
| p-value | | 0.130 | | 0.196 | | 0.122 | | 0.704 | |

*significant p-value <0.05; IP-In-patient; OP-Out-patient; SD Standard deviation.

**Table 7. Backward multiple linear regression of significant factors associated with WHOQOL-BREF.**

| Domains | Variables | Unstandardized coefficients | | Standardized coefficients | t | p-value |
|---------|-----------|------|------|------|------|------|
| | | B | SE | Beta | | |
| Domain 3 | Gender | -3.369 | 1.357 | -0.141 | -2.483 | **0.014***  |
| | Medication adherence | 1.950 | 1.008 | 0.110 | 1.935 | **0.050***  |

*Significant p-value <0.05.

The Quality of Life (WHOQOL-BREF) domains assessed were physical health, psychological health, social relationships), and environment). Among these domains, social relationships had the highest satisfaction score (mean: 64.91), while physical health had the lowest satisfaction score (mean: 41.83). The findings of the current study, where social relationships ranked as the domain with the highest satisfaction score and physical health had the lowest, are consistent with some previous research [38–40]. Social relationships have been consistently identified as a significant contributor to overall quality of life, particularly among older adults. However, variations in domain satisfaction scores can be influenced by cultural and demographic factors. Comparing these findings to other studies provides a broader perspective on the relationship between domain satisfaction and overall quality of life. The Morisky medication adherence scale (MMAS-8) was employed to assess medication adherence. The findings of MMAS-8 revealed that a significant proportion of participants exhibited medium adherence (47.1%), followed by low adherence (42.3%), and high adherence (10.6%). Previous studies are consistent with these findings [41–43]. In contrast to our study findings, Punnapurath *et al.* [44] conducted a study in which the level of adherence was reported as high in 82% of participants, medium in 16%, and low in 2%. The findings of medium, low, and high adherence levels among elderly patients emphasize the need for targeted interventions and support to improve medication adherence. This is especially important in this demographic, given the unique challenges they may face in managing their medications. Understanding and addressing factors contributing to medication non-adherence among the elderly can have a significant impact on their overall well-being and the effectiveness of their medical treatments.

The statistically significant correlation was observed in our study between medication adherence and social relationships of the WHOQOL-BREF score aligns with the previous research [45–47], highlighting the importance of social connections in influencing medication adherence and, by extension, quality of life among elderly patients. Prior research has indicated that anxiety and sadness play significant roles in the quality of life of individuals living with chronic illnesses [46]. In their study, Cohen [48] observed a significant association between quality of life and social support among individuals diagnosed with Huntington's disease. Untas *et al.* [49] proposed that a correlation exists between inadequate social support and increased mortality risk, reduced adherence to medical treatment, and diminished physical quality of life among those with chronic medical conditions. The findings imply that an individual's social relationships, including interactions with family, friends, and the community, play a role in their ability and willingness to adhere to their prescribed medication regimens. Strong social support networks can positively influence a person's motivation to follow their treatment plan.

The WHOQOL-BREF is designed to provide a comprehensive assessment of an individual's quality of life. It considers physical, psychological, social, and environmental factors, acknowledging that these domains are not isolated but influence each other [50]. Our study finding revealed a statistically significant correlation between all four domains of the

WHOQOL-BREF score. Similar findings were reported in previous studies [38,51–54]. Clinicians and healthcare providers often use quality of life assessments to understand the impact of medical conditions and treatments on patients [55]. The observed correlations among domains emphasize the importance of addressing multiple aspects of a patient's well-being to achieve holistic care. Agborsangaya *et al.* [56] support the idea that considering multiple quality-of-life domains can guide clinical decision-making. This finding reflects the idea that individuals' well-being is influenced by various interconnected aspects of their lives and assessing these domains comprehensively can provide valuable insights for healthcare planning and interventions.

This study revealed statistically significant differences in mean scores across various domains of the WHOQOL-BREF (all pairs 2, 3, 4, 5, and 6) except the pair 1 (DOM1-DOM2) health domains. These findings agree with the previous studies among different populations [54,57–60]. In contrast, there were no significant variations identified across all domains in other studies [61,62]. The lack of a significant difference between physical health and psychological health scores might indicate that, in this particular study population, these two domains are closely related or influenced by similar factors. Research has shown that physical health can have a substantial impact on psychological well-being and vice versa. For example, individuals with chronic physical conditions may experience psychological distress [63]. These findings highlight the multidimensional nature of quality of life and suggest potential areas for targeted interventions and healthcare planning.

The study revealed that, on average, males reported higher satisfaction ratings in all four domains of the WHOQOL-BREF compared to females. A statistically significant difference is observed about gender and Domain 3 of the WHOQOL-BREF score. Gender differences in the perception of quality of life have been documented in various studies [45,64–66]. A study by Choo *et al.* [67] found that gender played a significant role in how individuals assessed their quality of life, with males reporting higher scores in several domains. The higher mean and percentage of satisfaction ratings among males in the physical health domain (DOM1) may indicate that, in this particular study population, males perceived themselves as having better physical health. This could be due to differences in health behaviors, access to healthcare, or reporting biases between genders. Similarly, the higher satisfaction ratings in the psychological health domain (DOM2) among males might suggest that they reported better psychological well-being. However, it's essential to consider that cultural and societal factors can influence how individuals, especially men, express their emotional and psychological states [68]. The finding of greater satisfaction ratings among males in the social relationships domain (DOM3) may be noteworthy. It could imply that, in this specific population, males felt more satisfied with their social interactions, social support, and relationships. This might reflect differences in the quality and nature of social relationships between genders [69]. The higher mean and percentage of satisfaction ratings in the environmental domain (DOM4) among males could suggest that they perceived their living conditions, access to resources, and opportunities more positively compared to females. Social and economic factors may contribute to these differences [51].

The Backward Multiple Linear Regression analysis further confirmed the significance of gender and medication adherence (MMAS-8) in influencing the total WHOQOL score. This underscores the multifaceted nature of quality of life and the importance of both gender and medication adherence in determining overall well-being among the elderly. Medication adherence has been recognized as a crucial factor in achieving positive health outcomes and improving the quality of life, particularly among individuals with chronic conditions [70]. Many individuals, especially elderly patients, manage chronic conditions that require consistent medication adherence. Non-adherence can lead to worsened health and decreased quality of

life [71]. Gender can influence the availability and nature of social support networks. The quality and quantity of social relationships are linked to psychological and emotional well-being [72].

The present study's findings highlight the complex and interrelated factors that influence the quality of life and medication adherence among elderly individuals. We've illuminated the significance of addressing medication adherence issues, understanding gender-specific disparities in quality of life, and recognizing the multifaceted nature of well-being. These insights have practical implications for healthcare providers and policymakers, emphasizing the need for tailored interventions that consider the unique needs and challenges faced by elderly patients. By adopting a holistic, patient-centered approach that integrates medical care with strategies to enhance medication adherence and promote social support, we can aspire to enhance the quality of life of our ageing population. This study underscores the importance of continued research and proactive efforts to ensure the optimal health and happiness of our elderly citizens.

While our study primarily focused on assessing the correlation between medication adherence and QOL domains, we acknowledge the importance of exploring underlying factors contributing to medication non-adherence. These factors may include socio-demographic characteristics, health beliefs, social support networks, access to healthcare services, and individual perceptions of medication necessity and concerns. Understanding the interplay between these factors and medication adherence is essential for developing targeted interventions aimed at improving both medication adherence and overall QOL. By identifying barriers to medication adherence and addressing them through tailored interventions, healthcare providers can effectively support elderly patients in managing their medications and enhancing their well-being. Furthermore, integrating strategies to promote medication adherence into comprehensive healthcare plans can lead to better health outcomes and improved QOL for elderly individuals living with chronic conditions.

## Limitations

The study was conducted in a single medical college and hospital, which may not fully represent the diverse elderly population. Participants were limited to those seeking care at this specific institution, potentially introducing selection bias, and limiting the generalizability of the findings. The study design is cross-sectional, which means data was collected at a single point in time. This hinders the chances to establish differences of assessment over time. Longitudinal studies would be beneficial for understanding how medication adherence and quality of life evolve in the elderly. Quality of life is inherently subjective, and its assessment can be influenced by personal perceptions and cultural norms. While the WHOQOL-BREF is a validated tool, individual interpretations may vary.

## Conclusion

In conclusion, the study highlights the intricate relationship between medication adherence and quality of life among elderly patients, shedding light on gender differences in QOL perceptions. The significant association observed between medication adherence (assessed using MMAS-8) and QOL (evaluated using WHOQOL-BREF) underscores the importance of addressing medication adherence issues to enhance overall well-being in this vulnerable population. Moreover, the notable gender disparities in QOL ratings underscore the necessity for gender-sensitive healthcare interventions for the elderly. By considering these insights, healthcare providers can develop tailored interventions that address medication adherence challenges and improve QOL outcomes among elderly patients. However, further research is

warranted to explore specific factors influencing medication adherence and QOL in this demographic, and longitudinal studies can provide deeper insights into effective strategies for enhancing both medication adherence and QOL. Through ongoing research and tailored interventions, holistic care and well-being of elderly individuals can be improved.

## Supporting information

**S1 Data.**
(SAV)

## Acknowledgments

The authors acknowledge the medical staff and geriatric patients of Yenepoya Medical College and Hospital and Yenepoya Pharmacy college & Research Centre for providing valuable cooperation to complete this study successfully.

## Author Contributions

**Conceptualization:** Umaima Farheen Khaiser, Rokeya Sultana.

**Data curation:** Rokeya Sultana, Ranajit Das.

**Formal analysis:** Umaima Farheen Khaiser, Rokeya Sultana, Ranajit Das.

**Funding acquisition:** Saeed G. Alzahrani, Shahabe Saquib, Shaheen Shamsuddin.

**Investigation:** Umaima Farheen Khaiser, Rokeya Sultana.

**Methodology:** Umaima Farheen Khaiser, Rokeya Sultana, Mohammad Fareed.

**Resources:** Umaima Farheen Khaiser.

**Software:** Ranajit Das.

**Supervision:** Rokeya Sultana.

**Validation:** Rokeya Sultana, Ranajit Das, Saeed G. Alzahrani, Mohammad Fareed.

**Visualization:** Umaima Farheen Khaiser, Saeed G. Alzahrani.

**Writing – original draft:** Umaima Farheen Khaiser, Rokeya Sultana.

**Writing – review & editing:** Umaima Farheen Khaiser, Rokeya Sultana, Saeed G. Alzahrani, Shahabe Saquib, Shaheen Shamsuddin, Mohammad Fareed.

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
